# Inhibition Studies on Human and Mycobacterial Carbonic Anhydrases with *N*-((4-Sulfamoylphenyl)carbamothioyl) Amides

**DOI:** 10.3390/molecules28104020

**Published:** 2023-05-11

**Authors:** Morteza Abdoli, Alessandro Bonardi, Niccolò Paoletti, Ashok Aspatwar, Seppo Parkkila, Paola Gratteri, Claudiu T. Supuran, Raivis Žalubovskis

**Affiliations:** 1Institute of Technology of Organic Chemistry, Faculty of Materials Science and Applied Chemistry, Riga Technical University, LV-1048 Riga, Latvia; 2Neurofarba Department, Universitàdegli Studi di Firenze, 50019 Florence, Italyniccolo.paoletti@unifi.it (N.P.); 3Laboratory of Molecular Modeling Cheminformatics & QSAR, Neurofarba Department, Università degli Studi di Firenze, 50019 Florence, Italy; paola.gratteri@unifi.it; 4Faculty of Medicine and Health Technology, Tampere University, 33520 Tampere, Finland; 5Fimlab Ltd., Tampere University Hospital, 33520 Tampere, Finland; 6Latvian Institute of Organic Synthesis, LV-1006 Riga, Latvia

**Keywords:** carbonic anhydrase, inhibitors, sulfonamides, bacterial enzymes, *Mycobacterium tuberculosis*, in silico studies

## Abstract

A library of structurally diverse *N*-((4-sulfamoylphenyl)carbamothioyl) amides was synthesized by selective acylation of easily accessible 4-thioureidobenzenesulfonamide with various aliphatic, benzylic, vinylic and aromatic acyl chlorides under mild conditions. Inhibition of three α-class cytosolic human (h) carbonic anhydrases (CAs) (EC 4.2.1.1); that is, hCA I, hCA II and hCA VII and three bacterial β-CAs from *Mycobacterium tuberculosis* (MtCA1-MtCA3) with these sulfonamides was thereafter investigated in vitro and in silico. Many of the evaluated compounds displayed better inhibition against hCA I (K_I_ = 13.3–87.6 nM), hCA II (K_I_ = 5.3–384.3 nM), and hCA VII (K_I_ = 1.1–13.5 nM) compared with acetazolamide (AAZ) as the control drug (K_I_ values of 250, 12.5 and 2.5 nM, respectively, against hCA I, hCA II and hCA VII). The mycobacterial enzymes MtCA1 and MtCA2 were also effectively inhibited by these compounds. MtCA3 was, on the other hand, poorly inhibited by the sulfonamides reported here. The most sensitive mycobacterial enzyme to these inhibitors was MtCA2 in which 10 of the 12 evaluated compounds showed K_I_s (K_I_, the inhibitor constant) in the low nanomolar range.

## 1. Introduction

Carbonic anhydrases (CAs) (EC 4.2.1.1) constitute a superfamily of lyases, a class of enzymes that catalyze the interconversion between CO_2_ and water to bicarbonate and a proton [1], a simple but fundamental reaction that contributes to several important pathophysiological processes connected to pH buffering, metabolism, signaling and other processes [2]. According to their amino acid sequence similarity, eight genetically distinct families of CAs were described so far, including α, β, γ, δ, ζ, η, θ, and ι [3]. Human CAs (hCAs) belong to the α-class, which express 15 isoforms with different catalytic activity, tissue expression and subcellular localization [4]. Of these, twelve are catalytically active (CA I–IV, VA–VB, VI–VII, IX and XII–XIV), whereas three are inactive (CAs VIII, X and XI) [5]. It is well documented that abnormal expression and/or activities of CA isoforms are linked with various diseases, including glaucoma, epileptic seizures, obesity, cancer, neurological disorders, etc. [6]. Therefore, the isoform-selective inhibition of these isoforms is found in pharmacological applications in many areas.

Since the early 1940s when Mann and Keilin discovered that sulfonamides are potent inhibitors of CAs [7], a huge number of sulfonamide-based CAIs have been reported [8] and, as a result, many drugs containing a primary sulfonamide moiety are currently available on the market for the treatment of various diseases (Figure 1a) [9,10]. However, because hCA isoforms have a high degree of homology, clinical drugs designed to inhibit the enzymatic activity of a particular isoform can also bind to others with similar affinity, thus causing many undesirable side effects [11]. Therefore, many researchers around the world have been working to explore novel sulfonamide compounds with maximum inhibitory action against a particular isoform(s). In this context, over the last decade, several research groups have disclosed that sulfonamide–acyl thiourea derivatives were effective inhibitors of CAs (Figure 1b) [12,13,14,15,16]. In order to extend these efforts and in continuation of our interest in the study of sulfonamide CAIs [17,18,19,20,21,22,23,24,25,26,27,28,29,30,31,32,33,34,35,36], in the present study, we synthesized a panel of 12 structurally diverse *N*-((4-sulfamoylphenyl)carbamothioyl) amides by the reaction of easily available 4-thioureidobenzenesulfonamide with the appropriate acid chlorides and investigated their inhibitory activities against three human CAs (hCA I, hCA II and hCA VII) and three bacterial β-CAs from *Mycobacterium tuberculosis* (MtCA1-MtCA3), which were recently validated as effective targets for the development of antituberculosis agents [37,38,39,40,41,42,43], to discover possible promising drug candidate(s).

## 2. Results and Discussion

### 2.1. Chemistry

A series of *N*-((4-sulfamoylphenyl)carbamothioyl) amide derivatives were synthesized according to the general Figure 1. All compounds were thoroughly characterized by HRMS, ^1^H NMR and ^13^C NMR which proved their structure (see Section 4 for details). 4-Thioureidobenzenesulfonamide **1** was synthesized according to previously reported procedures [17,18,19,20,21], and then it was subjected to acylation with various aliphatic, vinylic and aromatic acid chlorides of **2** to providethe desired compounds of **3** in satisfactory yields, ranging from 22% to 75%. Notably, the yields corresponded to the *N*-acylation step, not the overall yield.

### 2.2. Carbonic Anhydrase Inhibition

The *N*-((4-sulfamoylphenyl)carbamothioyl) amides **3a**–**l** reported here were tested as inhibitors of three cytosolic hCA isoforms (hCA I, II and VII), as well as three bacterial β-CAs from *M. tuberculosis* (MtCA1-MtCA3) using a stopped-flow CO_2_ hydrase assay [44]. The classical sulfonamide CAI acetazolamide (5-acetamido-1,3,4-thiadiazole-2-sulfonamide (**AAZ**)) was used as the reference drug for the measurements reported in Table 1.

The analysis and interpretation of data presented in Table 1 led to the following structure–activity relationships (SARs):(i)The physiologically dominant cytosolic human isoform hCA I was effectively inhibited by all evaluated sulfonamides **3a**–**l**, with K_I_s in the range of 13.3–87.6 nM. As shown in Table 1, all tested compounds exhibited better inhibition against this isozyme compared with acetazolamide (**AAZ**) as the control drug. Interestingly, the poorest inhibitor of the series, **3l**, inhibited hCA I up to three-fold compared to the reference inhibitor (K_I_ = 250 nM). Since the structure–activity relationship (SAR) was not flat, with the aim of simplifying its interpretation, the compounds were classified into five classes: (i) acyclic alkyl-; (ii) cyclic alkyl-; (iii) benzyl-; (iv) vinyl-; and (iv) aryl-substituted derivatives. The SARs for the linear alkyl-substituted compounds **3a**–**c** and **3e**–**g** indicated that a higher alkyl chain length leads to more potent inhibitory activities against hCA I. Therefore, hexanamide **3g** exhibited the most engaging activity, with a K_I_ value of 13.3 nM. Similarly, by the expansion of the ring size of the cyclic alkyl-substituted derivatives **3d**,**h**, their inhibitory potency increased; albeit there is no clear relationship between the inhibitory activity of acyclic and cyclic series. On the other hand, the benzyl- and vinyl-substituted derivatives **3i** and **3j,k**, respectively, exhibited similar inhibitory activities, which were almost equal to the medium-sized ring alkane-substituted **3l**. Finally, among the investigated compounds, the only aryl-substituted derivative, **3l**, exhibited the poorest inhibition for this isozyme. In summary, the potency for the inhibition of hCA I by the newly designed compounds followed the order: long aliphatic chain (C6)-substituted > medium aliphatic chain (C2-C4)-substituted ≥ medium-sized cyclic aliphatic (C6)-substituted ≈ vinyl-substituted ≈ benzyl-substituted > small-sized cyclic aliphatic (C3)-substituted ≈ aryl-substituted derivatives.(ii)The physiologically most relevant and fastest isoform, hCA II, was also effectively inhibited by most of the evaluated sulfonamides, with K_I_s ranging between 5.3 and 384.3 nM. Notably, half of compounds reported here displayed better inhibitory activity towards hCA II compared to **AAZ**. Again, a linear alkyl-substituted sulfonamide, **3f**, showed superior activity, with a K_I_ value of 5.3 nM, which was 2.5-fold higher than that of **AAZ**. The SAR was rather similar to that outlined above for hCA I, with the most obvious difference being in the case of the aryl-substituted derivatives. While for hCA I the worst inhibition was observed for compound **3l**, and this compound showed one of the best inhibition values of the series against hCA II (K_I_ of 6.8 nM). Needless to say that the high similarity of the observed SARs for hCA I and hCA II can be explained by the high-sequence homology of the amino acid present within the active site of these isozymes [45].(iii)The other cytosolic isoform, hCA VII, which was recently validated as a therapeutic target in neuropathic pain [46], was also strongly inhibited by all evaluated compounds (K_I_s in the range of 1.1–13.5 nM) compared to **AAZ** with a K_I_ of 2.5 nM. The data presented in Table 1 indicate that one-third of the compounds investigated here (**3d**–**f** and **3h**) displayed even better inhibitory activities against hCA VII in comparison with **AAZ**. Among them, **3d** showed superior selectivity against this isoform versus hCA I and hCA II, which was more than 46 and 202 times more selective against hCA VII vs. hCA I and hCA II, respectively. Therefore, this compound may be considered an interesting starting point for the development of hCA VII-selective inhibitors, which may be used as neuropathic-attenuating agents.(iv)As seen from the data in Table 1, the tested sulfonamides exhibited good to moderate inhibitory action against the mycobacterial enzyme MtCA1, with nanomolar to micromolar efficacies (K_I_s of 95.2 nM to 6.669 μM). The SAR is diverse from what was observed for the α-isoforms, except for the most massive aliphatic (C6)-substituted derivatives **3g**,**h**, the rest of the studied derivatives were active in the nanomolar range. Compound **3d**, which exhibited the weakest results for hCA I and II, displayed the best activity against MtCA1, with an K_I_ value of 95.2 nM andfive-fold superior to acetazolamide (K_I_ of 480 nM). The results are highly encouraging towards their future use in designing β-CA-selective inhibitors.(v)The second *M. tuberculosis* isoform, MtCA2, was the best inhibited bacterial CA among the three such enzymes investigated in this study, with *N*-((4-sulfamoylphenyl)carbamothioyl) amides **3a**–**l**. Indeed, all of these compounds showed low nanomolar inhibition constants ranging between 3.4 and 57.1 nM. It is worthwhile to note that almost 85% of the tested compounds **3a**–**d** and **3g**–**l** displayed better inhibitory activities than **AAZ**. This means that all of the substitution patterns explored here led to highly effective MtCA2 inhibitors.(vi)MtCA3 was, on the other hand, less sensitive to inhibition with the evaluated compounds compared to MtCA1 and MtCA2, and the K_I_s were in the range of 446.6–9396 nM. In this case, the styrene-substituted derivative **3k** demonstrated the best activity, with a K_I_ value of 446.6 nM but was still 4.3-fold less potent than acetazolamide (K_I_ of 104 nM).

## 3. In Silico Studies

In silico studies were performed to investigate indepth the binding properties of the best N-((4-sulfamoylphenyl)carbamothioyl) amides **3d** and **3k** towards the human α-CA I, II and VII and against the *Mycobacterium tuberculosis* isoforms MtCA1, MtCA2 and MtCA3, belonging to the β-class. 

Because the 3D-solved structure of the MtCA3 isoform is not available to date, the homology model (HM) was developed using the solved coordinates of the “open” form of the β-CA from *Synechocystis* sp. PCC 6803 as a template (PDB 5SWC) [47]. The template shows the highest sequence identity percentage (34.1%) with the target (Appendix A). Docking was carried out using the best-scored model among the ones built by homology. It is noteworthy that, to the best of our knowledge, neither structural nor modeling studies have been reported to date on inhibitors of the MtCA3 isoform. All docking solutions obtained with the studied human CAs found the benzenesulfonamide bound to the zinc ion of the active site with the deprotonated nitrogen atom of the sulfonamide moiety (SO_2_NH^−^), completing the tetrahedral coordination sphere of the metal (Figure 2). Moreover, the benzenesulfonamide binding mode was stabilized by two H-bonds formed between the sulfonamide S=O and NH^−^ groups with a backbone NH and side-chain OH of T199, respectively. The stabilization was further strengthened by van der Waals contacts (vdW) established by the aromatic ring with residues A121/V121 (CA I/CA II and CA VII), V143, L198 and W205.

In CA I, the C=O group of cyclopropane carboxamide pendant of **3d** is an H-bond distance with the side-chain NH_2_ of Q92, a highly conserved residue of the hydrophilic half of the hCAs’ active site. Interestingly, the longer cinnamic tail of **3k** is oriented towards the lipophilic part of the active site giving rise to vdW interactions with residues L131, A135, V207 and P202 and T-shaped π-π stacking contacts with the aromatic ring of the peculiar Y204 residue side chain (Figure 2A). This wide network of interactions could provide an explanation for the better CA I inhibitory activity of **3k** than **3d**. 

In the active site of CA II, the C=S group of ligand **3d** engages in an H-bond with the side-chain NH_2_ of Q92 and the cyclopropane ring lodges in the center of the active site, interacting with F131. The C=O group of the cinnamide pendant of **3k** is H-bonded with the side-chain OH of T200. It is likely that the contribution to the stabilization of the docking pose given by the van der Waals contacts with P202, P201 and W5, coupled with the steric hindrance exerted by the proximity of the cinnamide pendant to the proton shuttle H64 residue, contributes to the better CA II K**_I_** value of **3k** versus **3d** (Figure 2B).

In CA VII, the C=O groups of both **3d** and **3k** engage in H-bonds with the side-chain NH_2_ of Q92 and Q67, respectively, placing the respective pendants towards K91, a peculiar residue of this isozyme. In particular, the cyclopropane ring of **3d** interacts with F131 and the side-chain carbons of K91, while the aromatic ring of the longer cinnamic tail of **3k** engages in π-cation interactions with the K91 side-chain NH_3_^+^. These interactions greatly influence the positioning of the pendant. However, the pose is also well destabilized by the proximity of the aromatic ring and the negatively charged sidechain of E69, which could explain the difference in the K_I_ values observed for the derivatives **3k** and **3d** (Figure 2C).

All the β-CAs are characterized by a dimeric active site. Each of the *M. tuberculosis* CA isoforms (MtCA1, MtCA2 and MtCA3) has peculiar structural motifs that are between residues 91 and 109, 107 and 143, and 645 and 680 of MtCA1, MtCA2 and MtCA3-HM, respectively (Figure 3). The area of the enzymes resulting from the different random coil and α-helix patterns accommodates the tails of **3d** and **3k**. In all three enzyme isoforms, the pendants are stabilized by a series of van der Waals interactions, while the tetrahedral coordination sphere around the zinc ion is completed by the negatively charged nitrogen atom of the benzenesulfonamide moiety. In addition, an H-bond is established between the sulfonamide NH^−^ and the side-chain COO- of D37/D53/D586 (MtCA1/MtCA2/MtCA3-HM), the residue of the highly conserved dyad Asp-Arg responsible for the form “open”/“closed” pH-dependent interconversion in the β-CAs. Other H-bonds involving the sulfonamide moiety are formed with G92 (MtCA1, N-H…NH_2_), Q42 (MtCA2, S=O…NH_2_), Q575(MtCA3-HM, S=O…NH_2_) Y603 (MtCA3-HM, S=O…H-O) and G609 (MtCA3-HM, S=O…H-N). The aromatic ring of benzenesulfonamide is stabilized by vdW interactions with A59, G92, M93, F96, M24, I73, L77 and L78 in MtCA1, with A75, G108, A109, A112 and Y89 in MtCA2 and with L608, A646, A647, A650, M663 A109 and F627 in MtCA3-HM. In the last two isoenzymes, π-π stacking with Y89 (MtCA2) and F627 (MtCA3-HM) is also present (Figure 3).

Of relevance in MtCA1 is the H-bond formation between the side-chain T95 OH and the amidic NH (calculated pKa = 8.01) of the cyclopropane carboxamide tail of **3d** (Figure 3A). The calculated pKa of the amidic NH of ligand **3k** is 6.67; the deprotonated form present at physiological pH = 7.2–7.4 is not able to engage an H-bond as a donor with T95, thus explaining the worst inhibition profile of **3k**.

In MtCA2 active site, both ligands are stabilized by a wide network of hydrophobic interactions with A112, A115 (**3k**), T121(**3k**) and P123 (Figure 3B). The stronger benzenesulfonamide interaction within the MtCA2 active site joined with the hydrophobic nature of the ligand pendants and the target counterpart resulted in better K_I_ values versus MtCA1. 

Along with the prevalence of hydrophobic residues, the corresponding pocket of the homology-built model of MtCA3 also contains hydrophilic residues, which probably makes the interaction of **3d** and **3k** suboptimal with respect to the other MtCAs (Figure 3C). Indeed, the hydrophobic pendant of **3d** is oriented towards the side-chain COO- of E653, while the aromatic ring of the cinnamic tail of **3k**, achieving the complementary hydrophobic residues P656 and A657, is closed to the hydrophilic residues E653, T659 and T660.

## 4. Materials and Methods

### 4.1. Chemistry

Unlike 4-thioureidobenzenesulfonamide, which was prepared according to procedures in the literature, the other reagents, solvents and starting materials were obtained from commercial sources and were used as received without further purification. Thin-layer chromatography (TLC) was performed on silica gel, and spots were visualized with UV light (254 and 365 nm). The ^1^H and ^13^C NMR spectra were recorded with 500 and 125 MHz, respectively, using Bruker Avance instrument in DMSO-d_6_ with chemical shifts values (*δ*) in ppm relative to tetramethylsilane (TMS). High-resolution mass spectra (HRMS) were recorded on a mass spectrometer with a Q-TOF micro mass analyzer using the ESI technique.

### 4.2. Synthesis

#### 4.2.1. Synthesis of *N*-((4-Sulfamoylphenyl)carbamothioyl)acetamide **3a**

To a solution of 4-thioureidobenzenesulfonamide (**1**) (0.5 g, 2.16 mmol, 1.0 equiv.) and NaI (0.648 g, 4.32 mmol, 2.0 equiv.) in dry DMF (5 mL) at 0 °C, acetyl chloride (**2a**) (0.154 mL, 2.16 mmol, 1.0 equiv.) was slowly added while stirring. The reaction was allowed to warm to 20 °C and then stirred for 1 h. The reaction mixture was then treated with water (80 mL) and Et_2_O (10 mL), and the solution was vigorously stirred for 3 h. The solids formed in the organic phase were filtered and washed with water (30 mL) and Et_2_O (20 mL) to afford **3a** in 28% yield (168 mg) as a white powder.



^1^H NMR (500 MHz, DMSO-d_6_) δ = 2.20 (s, 3H), 7.42 (s, 2H), 7.45–7.91 (m, 4H), 11.63 (s, 1H), 12.67 (s, 1H) ppm ^13^C NMR (125 MHz, DMSO-d_6_) δ = 24.8, 125.2, 127.2, 141.6, 142.2, 173.8, 180.0 ppm HRMS (ESI) [M + H]^+^: *m*/*z* calcd. for (C_9_H_12_N_3_O_3_S_2_) 274.0320. Found: 274.0322.

#### 4.2.2. Synthesis of *N*-((4-Sulfamoylphenyl)carbamothioyl)propionamide **3b**

To a solution of 4-thioureidobenzenesulfonamide (**1**) (0.5 g, 2.16 mmol, 1.0 equiv.) and NaI (0.648 g, 4.32 mmol, 2.0 equiv.) in dry DMF (5 mL) at 0 °C, propionyl chloride (**2b**) (0.188 mL, 2.16 mmol, 1.0 equiv.) was slowly added with stirring. The reaction was allowed to warm to 20 °C and then stirred for 15 min. The reaction mixture was then treated with water (100 mL) and the solids formed were filtered and washed with water (30 mL) and DCM (20 mL) to afford **3b** in a 51% yield (317 mg) as a white powder.



^1^H NMR (500 MHz, DMSO-d_6_) δ = 0.96 (t, 3H, *J* = 6.8 Hz), 2.15 (q, 2H, *J* = 6.8 Hz), 7.51 (s, 2H), 7.54 (d, 2H, *J* = 7.8 Hz), 7.89 (d, 2H, *J* = 7.8 Hz), 9.88 (s, 1H), 10.27 (s, 1H) ppm ^13^C NMR (125 MHz, DMSO-d_6_) δ = 9.7, 32.3, 127.2, 131.3, 144.5, 145.0, 176.6, 185.9 ppm HRMS (ESI) [M + H]^+^: *m*/*z* calcd. for (C_10_H_14_N_3_O_3_S_2_) 288.0477. Found: 288.0478.

#### 4.2.3. Synthesis of *N*-((4-Sulfamoylphenyl)carbamothioyl)butyramide **3c**

To a solution of 4-thioureidobenzenesulfonamide (**1**) (0.5 g, 2.16 mmol, 1.0 equiv.) and NaI (0.648 g, 4.32 mmol, 2.0 equiv.) in dry DMF (5 mL) at 0 °C butyryl chloride (**2c**) (0.223 mL, 2.16 mmol, 1.0 equiv.) was slowly added with stirring. The reaction was allowed to warm to 20 °C and then stirred for 12 h. The reaction mixture was then treated with water (100 mL) and the solids formed were filtered and washed with water (30 mL) and DCM (20 mL) to afford **3c** in 34% yield (220 mg) as a light yellow powder.



^1^H NMR (500 MHz, DMSO-d_6_) δ = 0.95 (t, 3H, *J* = 7.0 Hz), 1.59–1.67 (m, 2H), 1.63 (t, 2H, *J* = 6.8 Hz), 7.41 (s, 2H), 7.86 (d, 2H, *J* = 8.1 Hz), 7.90 (d, 2H, *J* = 8.1 Hz), 11.59 (s, 1H), 12.72 (s, 1H) ppm ^13^C NMR (125 MHz, DMSO-d_6_) δ = 14.4, 18.7, 38.6, 125.2, 127.2, 141.6, 142.2, 176.4, 180.0 ppm HRMS (ESI) [M + H]^+^: *m*/*z* calcd. for (C_11_H_16_N_3_O_3_S_2_) 302.0633. Found: 302.0640.

#### 4.2.4. Synthesis of *N*-((4-Sulfamoylphenyl)carbamothioyl)cyclopropanecarboxamide **3d**

To a solution of 4-thioureidobenzenesulfonamide (**1**) (0.5 g, 2.16 mmol, 1.0 equiv.) and NaI (0.648 g, 4.32 mmol, 2.0 equiv.) in dry DMF (5 mL) at 0 °C cyclopropanecarbonyl chloride (**2d**) (0.196 mL, 2.16 mmol, 1.0 equiv.) was slowly added with stirring. The reaction was allowed to warm to 20 °C and stirred for 15 min. The reaction mixture was then treated with water (100 mL) and the solids formed were filtered and washed with water (30 mL) and DCM (20 mL) to afford **3d** in 75% yield (489 mg) as a white powder.



^1^H NMR (500 MHz, DMSO-d_6_) δ = 0.82–0.86 (m, 2H), 0.97–1.02 (m, 2H), 1.28–1.34 (m, 1H), 7.51 (s, 2H), 7.58 (d, 2H, *J* = 8.2 Hz), 7.91 (d, 2H, *J* = 8.2 Hz), 9.91 (s, 1H), 10.14 (s, 1H) ppm ^13^C NMR (125 MHz, DMSO-d_6_) δ = 11.5, 17.0, 127.4, 131.1, 144.4, 145.1, 175.9, 185.8 ppm HRMS (ESI) [M + H]^+^: *m*/*z* calcd. for (C_11_H_14_N_3_O_3_S_2_) 300.0477. Found: 300.0480.

#### 4.2.5. Synthesis of *N*-((4-sulfamoylphenyl)carbamothioyl)isobutyramide **3e**

To a solution of 4-thioureidobenzenesulfonamide (**1**) (0.5 g, 2.16 mmol, 1.0 equiv.) and NaI (0.648 g, 4.32 mmol, 2.0 equiv.) in dry DMF (5 mL) at 0 °C, isobutyryl chloride (**2e**) (0.226 mL, 2.16 mmol, 1.0 equiv.) was slowly added with stirring. The reaction was allowed to warm to 20 °C and then stirred for 1 h. The reaction mixture was then treated with water (100 mL) and the solids formed were filtered and washed with water (30 mL) and DCM (20 mL) to afford **3e** in 72% yield (469 mg) as a white powder.



^1^H NMR (500 MHz, DMSO-d_6_) δ = 1.48 (d, 6H, *J* = 6.2 Hz), 2.82–2.88 (m, 1H), 7.41 (s, 2H), 7.86 (d, 2H, *J* = 7.9 Hz), 7.90 (d, 2H, *J* = 7.9 Hz), 11.60 (s, 1H), 12.79 (s, 1H) ppm ^13^C NMR (125 MHz, DMSO-d_6_) δ = 19.8, 35.4, 125.2, 127.2, 141.6, 142.2, 180.3, 180.3 ppm HRMS (ESI) [M + H]^+^: *m*/*z* calcd.For (C_11_H_16_N_3_O_3_S_2_) 302.0633. Found: 302.0645.

#### 4.2.6. Synthesis of 3-methyl-*N*-((4-sulfamoylphenyl)carbamothioyl)butanamide **3f**

To a solution of 4-thioureidobenzenesulfonamide (**1**) (0.5 g, 2.16 mmol, 1.0 equiv.) and NaI (0.648 g, 4.32 mmol, 2.0 equiv.) in dry DMF (5 mL) at 0 °C, 3-methylbutanoyl chloride (**2f**) (0.264 mL, 2.16 mmol, 1.0 equiv.) was slowly added with stirring. The reaction was allowed to warm to 20 °C and then stirred for 2 h. The reaction mixture was then treated with water (100 mL) and the solids formed were filtered and washed with water (30 mL) and DCM (20 mL) to afford **3f** in 31% yield (210 mg) as a white powder.



^1^H NMR (500 MHz, DMSO-d_6_) δ = 0.93 (d, 6H, *J* = 6.6 Hz), 2.01–2.10 (m, 1H), 2.36 (d, 2H, *J* = 7.0 Hz), 7.39 (s, 2H), 7.82 (d, 2H, *J* = 8.7 Hz), 7.86 (d, 2H, *J* = 8.7 Hz), 11.57 (s, 1H), 12.69 (s, 1H) ppm ^13^C NMR (125 MHz, DMSO-d_6_) δ = 23.2, 25.7, 46.9, 127.3, 131.1, 144.4, 145.0, 174.8, 186.2 ppm HRMS (ESI) [M + H]^+^: *m*/*z* calcd. for (C_12_H_18_N_3_O_3_S_2_) 316.0790. Found: 316.0795.

#### 4.2.7. Synthesis of *N*-((4-sulfamoylphenyl)carbamothioyl)hexanamide **3g**

To a solution of 4-thioureidobenzenesulfonamide (**1**) (0.5 g, 2.16 mmol, 1.0 equiv.) and NaI (0.648 g, 4.32 mmol, 2.0 equiv.) in dry DMF (5 mL) at 0 °C, hexanoyl chloride (**2g**) (0.302 mL, 2.16 mmol, 1.0 equiv.) was slowly added with stirring. The reaction was allowed to warm to 20 °C and then stirred for 12 h. The reaction mixture was then treated with water (80 mL) and Et_2_O (10 mL) and the solution was vigorously stirred. The solids formed in the organic phase were filtered and washed with water (30 mL) and Et_2_O (20 mL), and DCM (20 mL) to afford **3g** in 35% yield (258 mg) as a light yellow powder.



^1^H NMR (500 MHz, DMSO-d_6_) δ = 0.91 (t, 3H, *J* = 6.9 Hz), 1.28–1.36 (m, 4H), 1.57–1.65 (m, 2H), 2.50 (t, 2H, *J* = 6.5 Hz), 7.41 (s, 2H), 7.86 (d, 2H, *J* = 7.4 Hz), 7.90 (d, 2H, *J* = 7.4 Hz), 11.59 (s, 1H), 12.72 (s, 1H) ppm ^13^C NMR (125 MHz, DMSO-d_6_) δ = 14.7, 22.8, 24.9, 31.6, 36.7, 125.1, 127.2, 141.6, 142.2, 176.6, 180.0 ppm HRMS (ESI) [M + H]^+^: *m*/*z* calcd. for (C_13_H_20_N_3_O_3_S_2_) 330.0946. Found: 330.0952.

#### 4.2.8. Synthesis of *N*-((4-Sulfamoylphenyl)carbamothioyl)cyclohexanecarboxamide **3h**

To a solution of 4-thioureidobenzenesulfonamide (**1**) (0.5 g, 2.16 mmol, 1.0 equiv.) and NaI (0.648 g, 4.32 mmol, 2.0 equiv.) in dry DMF (5 mL) at 0 °C, cyclohexanecarbonyl chloride (**2h**) (0.291 mL, 2.16 mmol, 1.0 equiv.) was slowly added with stirring. The reaction was allowed to warm to 20 °C and then stirred for 4 h. The reaction mixture was then treated with water (80 mL) and Et_2_O (10 mL), and the solution was vigorously stirred. The solids formed in the organic phase were filtered and washed with water (30 mL) and Et_2_O (20 mL), and DCM (20 mL) to afford **3h** in 29% yield (212 mg) as a white powder.



^1^H NMR (500 MHz, DMSO-d_6_) δ = 1.19–1.31 (m, 3H), 1.35–1.43 (m, 2H), 1.64–1.70 (m, 1H), 1.75–1.90 (m, 4H), 2.58–2.64 (m, 1H), 7.41 (s, 2H), 7.86 (d, 2H, *J* = 7.9 Hz), 7.90 (d, 2H, *J* = 7.9 Hz), 11.54 (s, 1H), 12.73 (s, 1H) ppm ^13^C NMR (125 MHz, DMSO-d_6_) δ = 25.9, 26.1, 29.5, 44.8, 125.1, 127.2, 141.6, 142.2, 179.3, 180.3 ppm HRMS (ESI) [M + H]^+^: *m*/*z* calcd. for (C_14_H_20_N_3_O_3_S_2_) 342.0946. Found: 342.0950.

#### 4.2.9. Synthesis of 2-(Naphthalen-1-yl)-*N*-((4-sulfamoylphenyl)carbamothioyl)acetamide **3i**

To a solution of 4-thioureidobenzenesulfonamide (**1**) (0.5 g, 2.16 mmol, 1.0 equiv.) and NaI (0.648 g, 4.32 mmol, 2.0 equiv.) in dry DMF (5 mL) at 0 °C, 2-(naphthalen-1-yl)acetyl chloride (**2i**) (0.443 mL, 2.16 mmol, 1.0 equiv.) was slowly added with stirring. The reaction was allowed to warm to 20 °C and then stirred for 12 h. The reaction mixture was then treated with water (80 mL) and Et_2_O (10 mL), and the solution was vigorously stirred. The solids formed in the organic phase were filtered and washed with water (30 mL) and *^t^*BuOMe (20 mL), and MeOH (20 mL) to afford **3i** in 22% yield (189 mg) as a white powder.



^1^H NMR (500 MHz, DMSO-d_6_) δ = 4.02 (s, 2H), 7.37–7.60 (m, 6H), 7.79–7.97 (m, 7H), 10.04 (s, 1H), 10.19 (s, 1H) ppm ^13^C NMR (125 MHz, DMSO-d_6_) δ = 42.7, 125.0, 126.3, 126.6, 127.3, 127.5, 128.6, 129.4, 131.1, 131.7, 132.8, 134.2, 144.7, 145.0, 173.4, 186.2 ppm HRMS (ESI) [M + H]^+^: *m*/*z* calcd. for (C_19_H_18_N_3_O_3_S_2_) 400.0790. Found: 400.0805.

#### 4.2.10. Synthesis of *N*-((4-Sulfamoylphenyl)carbamothioyl)but-2-enamide **3j**

To a solution of 4-thioureidobenzenesulfonamide (**1**) (0.5 g, 2.16 mmol, 1.0 equiv.) in dry DMF (5 mL) at 0 °C but-2-enoyl chloride (**2j**) (0.277 mL, 2.16 mmol, 1.0 equiv.) was slowly added with stirring. The reaction was allowed to warm to 20 °C and then stirred for 2 h. The reaction mixture was then treated with water (100 mL), and the solids formed were filtered and washed with water (30 mL) and *^i^*PrOH (20 mL) to afford **3j** in 46% yield (299 mg) as an orange powder.



^1^H NMR (500 MHz, DMSO-d_6_) δ = 1.94 (d, 3H, *J* = 5.7 Hz), 6.38 (d, 1H, *J* = 15.3 Hz), 7.05–7.13 (m, 1H), 7.42 (s, 2H), 7.87 (d, 2H, *J* = 8.0 Hz), 7.92 (d, 2H, *J* = 8.0 Hz), 11.62 (s, 1H), 12.88 (s, 1H) ppm ^13^C NMR (125 MHz, DMSO-d_6_) δ = 19.1, 124.7, 125.1, 127.2, 141.6, 142.2, 147.3, 167.0, 180.3 ppm HRMS (ESI) [M + H]^+^: *m*/*z* calcd. for (C_11_H_14_N_3_O_3_S_2_) 300.0477. Found: 300.0488.

#### 4.2.11. Synthesis of *N*-((4-Sulfamoylphenyl)carbamothioyl)cinnamamide **3k**

To a solution of 4-thioureidobenzenesulfonamide (**1**) (0.5 g, 2.16 mmol, 1.0 equiv.) in dry DMF (5 mL) at 0 °C cinnamoyl chloride (**2k**) (0.329 mL, 2.16 mmol, 1.0 equiv.) was slowly added with stirring. The reaction was allowed to warm to 20 °C and then stirred for 1 h. The reaction mixture was then treated with water (100 mL), and the solids formed were filtered and washed with water (30 mL) and DCM (20 mL) to afford **3k** in 31% yield (244 mg) as a light yellow powder.



^1^H NMR (500 MHz, DMSO-d_6_) δ = 6.45 (d, 1H, *J* = 15.4 Hz), 7.44–7.55 (m, 9H), 7.70 (d, 1H, *J* = 15.4 Hz), 7.93 (d, 2H, *J* = 6.5 Hz), 10.03 (s, 2H) ppm ^13^C NMR (125 MHz, DMSO-d_6_) δ = 120.8, 127.5, 129.1, 130.0, 130.6, 131.6, 135.0, 144.3, 144.5, 144.6, 167.4, 186.3 ppm HRMS (ESI) [M + H]^+^: *m*/*z* calcd. for (C_16_H_16_N_3_O_3_S_2_) 362.0633. Found: 362.0636.

#### 4.2.12. Synthesis of 4-nitro-*N*-((4-sulfamoylphenyl)carbamothioyl)benzamide **3l**

To a solution of 4-thioureidobenzenesulfonamide (**1**) (0.5 g, 2.16 mmol, 1.0 equiv.) in dry DMF (5 mL) at 0 °C 4-nitrobenzoyl chloride (**2l**) (0.401 g, 2.16 mmol, 1.0 equiv.) was slowly added with stirring. The reaction was allowed to warm to 20 °C and then stirred for 1 h. The reaction mixture was then treated with water (100 mL), and the solids formed were filtered and washed with water (30 mL) and Et_2_O (20 mL) to afford **3l** in 32% yield (267 mg) as a white powder.



^1^H NMR (500 MHz, DMSO-d_6_) δ = 7.47 (s, 2H), 7.52 (d, 2H, *J* = 7.9 Hz), 7.85 (d, 2H, *J* = 7.9 Hz), 7.89 (d, 2H, *J* = 8.1 Hz), 8.30 (d, 2H, *J* = 8.1 Hz), 9.66 (s, 1H), 10.06 (s, 1H) ppm ^13^C NMR (125 MHz, DMSO-d_6_) δ = 124.4, 127.5, 129.6, 130.2, 142.4, 143.8, 144.8, 149.5, 169.5, 187.1 ppm HRMS (ESI) [M + H]^+^: *m*/*z* calcd.For (C_14_H_13_N_4_O_5_S_2_) 381.0327. Found: 381.0330.

### 4.3. CA Inhibitory Assay

An applied photophysics stopped-flow instrument was used for assaying the CA catalyzed CO_2_ hydration activity [44]. Phenol red (at a concentration of 0.2 mM) was used as an indicator, working at an absorbance maximum of 557 nm, with 20 mM Hepes (pH 7.5) as buffer for α-CAs or 20 mM TRIS (pH 8.4) as a buffer for β-CAs and 20 mM Na_2_SO_4_ (for maintaining constant the ionic strength), following the initial rates of the CA-catalyzed CO_2_ hydration reaction for a period of 10–100 s. The CO_2_ concentrations ranged from 1.7 to 17 mM for the determination of the kinetic parameters and inhibition constants. For each inhibitor, at least six traces of the initial 5–10% of the reaction were used for determining the initial velocity. The unanalyzed rates were determined in the same manner and subtracted from the total observed rates. Stock solutions of inhibitor (0.1 mM) were prepared in distilled–deionized water and dilutions up to 0.01 nM were conducted thereafter with the assay buffer. Inhibitor and enzyme solutions were preincubated together for 6 h at room temperature prior to assay in order to allow for the formation of the E–I complex. The inhibition constants were obtained by nonlinear least squares methods using PRISM 3 and the Cheng–Prusoff equation, as reported earlier [48,49,50,51,52], and represent the mean from at least three different determinations. All CA isoforms were recombinant ones obtained in-house as reported earlier [53,54,55,56,57,58], and their concentrations in the assay system ranged between 7.6 and 12.5 nM.

### 4.4. In Silico Studies

The primary sequence of MtCA3 was retrieved from the UniProt Consortium. The crystal structure of β-CA from *Synechocystis* sp. PCC 6803 (PDB 5SWC; resolution 1.45 Å) [47] was used as a template in the homology modeling procedure (sequence alignment is reported in Appendix A). Multiple models were generated using the Prime module of Schrödinger [59] and the SwissModel platform (https://swissmodel.expasy.org/ (accessed on 29 March 2023)) [60] and submitted to loop refinements and quality evaluation procedures (Appendix A). The best-scored structure of MtCA3 and the crystal structure of CA I (PDB 2NMX) [61], CA II (PDB 3K34) [62], CA VII (PDB 6H38) [63], MtCA1 (PDB 1YLK) [43] and MtCA2 (PDB 2A5V) [42] downloaded by Protein Data Bank (RCSB.org) [64] were prepared using the Protein Preparation module implemented in Maestro Schrödinger suite [59,65,66,67,68,69], assigning bond orders, adding hydrogens, deleting water molecules and optimizing H-bonding networks. Finally, energy minimization with a root mean square deviation (RMSD) value of 0.30 was applied using an optimized potential for liquid simulation (OPLS4) force field [70,71,72,73,74]. The 3D ligand structures were prepared using Maestro [65] and evaluated for their ionization states at pH 7.3 ± 1.0 with Epik [66]. The conjugate gradient method in Macromodel [67] was used for energy minimization (maximum iteration number: 2500; convergence criterion: 0.05 Kcal/mol/Å2). The grids for docking were centered in the centroid of the complexed ligand. The docking studies were carried out with the program Glide [68] using the standard precision (SP) mode. 3D ligand structures were prepared using Maestro [65]. Figures were generated with Maestro and Chimera [59,65,66,67,68,69,75].

## 5. Conclusions

A panel of 12 structurally diverse *N*-((4-sulfamoylphenyl)carbamothioyl) amides were synthesized using selective acylation of easily available 4-thioureidobenzenesulfonamide with various aliphatic, benzylic, vinylic and aromatic acyl chlorides under mild conditions. The compounds were investigated as inhibitors of three human carbonic anhydrases (hCA I, hCA II and hCA VII) and three bacterial β-CAs from *Mycobacterium tuberculosis* (MtCA1-MtCA3). The results indicated that all targeted compounds had a higher inhibitory activity against hCA I than the standard drug, **AAZ**. On the other hand, 6 and 5 of the 12 evaluated compounds exhibited better or similar inhibition against hCA II and hCA VII, respectively, compared with **AAZ**. Of all the compounds investigated, **3d** exhibited superior selectivity against the brain-associated hCA VII isoform versus hCA I and hCA II, which was more than 46 and 202 times, respectively, more selective. Therefore, this compound may be considered an interesting starting point for the development of hCA VII-selective inhibitors which may be used as neuropathic-attenuating agents. In association with inhibition of bacterial β-CAs, the results showed that the assayed compounds were preferential inhibitors of MtCA2. Specially, compound **3g** displayed superior inhibitory activity with K_I_ values of 3.4 nM (three-fold higher than that of **AAZ**) and showed considerably effective selectivity of MtCA1/MtCA2 (>1033) and MtCA3/MtCA2 (>211). Therefore, this compound could be a potential and promising candidate for further investigations and in vivo experimentations for antibacterial drug discovery. Moreover, the binding mode of compounds **3d** and **3k** was investigated in silico in the active site of the six studied CA isoforms to unveil the relationship between the structural features and inhibition profiles. The absence of a 3D-solved structure of the MtCA3 β-CA required its homology model building, resulting in the first work that report structural information on this β-CA. 

## Data Availability

The authors confirm that the data supporting the findings of this study are available within the article and its Appendix A.

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
