# Peer review of "Inhibition Studies on Human and Mycobacterial Carbonic Anhydrases with *N*-((4-Sulfamoylphenyl)carbamothioyl) Amides"

_molecules, 2023, doi:10.3390/molecules28104020_

Round 1
Reviewer 1 Report
The manuscript entitled “Inhibition studies on human and mycobacterial carbonic anhy-drases with N-((4-sulfamoylphenyl)carbamothioyl) amides” deals with the synthesis 12 structurally diverse N-((4-sulfamoylphenyl)carbamothioyl) amides by selective acylation under mild conditions to test as inhibitors of three human carbonic anhydrases and three bacterial β- carbonic anhydrases from Mycobacterium tuberculosis. Carbonic anhydrases catalyze the interconversion between CO2 and water to bicarbonate and a proton, contributing to several important pathophysiological processes. The abnormal expression and/or activities of CA isoforms are linked with various diseases. Therefore, molecules that inhibit those enzymes can be good candidates to be used as pharmaceutical drugs. The authors showed interesting results with those synthesized compounds as inhibitors compared to a control drug. The innovation is the synthesis of these compounds that have not been tested to inhibit CA.
The manuscript is based on one result presented in Table 1, which is the inhibition of CA. I believe that the authors should explore the synthesis of the compounds, especially because the readers of Molecules would be interested on that subject. Explain (in the methodology) how the yields were calculated/measured and discuss those results.
Some other points should be addressed:
Abstract: “Many of the evaluated compounds displayed better inhibition against hCA I, hCA II, and hCA VII, respectively, compared with acetazolamide (AAZ) as control drug. The mycobacterial enzymes MtCA1 and MtCA2 were also effectively inhibited by these compounds.” It is important to show numbers. How much these compounds inhibited the enzymes?
Specify Kis in the abstract.
Which are the implications of those results?
Figure 1 legend: specify CAIs
Scheme 1: The percentual corresponds to what? Abbreviations should be specified.
Table 1: What Ki means? This must be clear in Table 1 without the need to read the methodology. I believe some parameters could be calculated with those results such as the inhibition in relation to the control, as a percentual. This would be good for the readers to understand the results immediately.
“The most important information of structure activity relationship (SAR) analysis is listed below.” Below, where? Please specify which Table or figure.
Discussion: Some information about the security of those synthesized molecules could be given.
Materials and Methods: The chemical formula should be presented as Figures and numbered accordingly.
Author Response
Comments and Suggestions for Authors
The manuscript entitled “Inhibition studies on human and mycobacterial carbonic anhy-drases with N-((4-sulfamoylphenyl)carbamothioyl) amides” deals with the synthesis 12 structurally diverse N-((4-sulfamoylphenyl)carbamothioyl) amides by selective acylation under mild conditions to test as inhibitors of three human carbonic anhydrases and three bacterial β- carbonic anhydrases from Mycobacterium tuberculosis. Carbonic anhydrases catalyze the interconversion between CO2 and water to bicarbonate and a proton, contributing to several important pathophysiological processes. The abnormal expression and/or activities of CA isoforms are linked with various diseases. Therefore, molecules that inhibit those enzymes can be good candidates to be used as pharmaceutical drugs. The authors showed interesting results with those synthesized compounds as inhibitors compared to a control drug. The innovation is the synthesis of these compounds that have not been tested to inhibit CA.
Q#1-The manuscript is based on one result presented in Table 1, which is the inhibition of CA. I believe that the authors should explore the synthesis of the compounds, especially because the readers of Molecules would be interested on that subject. Explain (in the methodology) how the yields were calculated/measured and discuss those results.
A#1- In accordance with your suggestion, the phrase "Notably, the yields correspond to the N-acylation step, not the overall yield." was added to the section 2.1. There is no need to say that the yields were calculated as follow:percent yield = actual yield/theoretical yield x 100.
Some other points should be addressed:
Q#2-Abstract: “Many of the evaluated compounds displayed better inhibition against hCA I, hCA II, and hCA VII, respectively, compared with acetazolamide (AAZ) as control drug. The mycobacterial enzymes MtCA1 and MtCA2 were also effectively inhibited by these compounds.” It is important to show numbers. How much these compounds inhibited the enzymes?
A#2-In accordance with your suggestion, the abstract was revised.
Q#3-Specify Kis in the abstract.
A#3-Done.
Q#4-Which are the implications of those results?
A#4- First of all the aim of present study was to expanding the database of carbonic anhydrase inhibitors. As highlighted in the conclusion part, owing to their high inhibitory activity and selectivity against particular isoforms, some of the tested compounds could be potential and promising candidates for further investigations and in vivo experimentations for antibacterial and anti-neuropathic drug discovery.
Q#5-Figure 1 legend: specify CAIs.
A#5-Done.
Q#6-Scheme 1: The percentual corresponds to what? Abbreviations should be specified.
A#6-The percentages in Scheme 1 represent the isolated yields.
Q#7-Table 1: What Ki means? This must be clear in Table 1 without the need to read the methodology. I believe some parameters could be calculated with those results such as the inhibition in relation to the control, as a percentual. This would be good for the readers to understand the results immediately.
A#7-We have specified the meaning of KI in the legend of the table as suggested by the reviewer.
Q#8-“The most important information of structure activity relationship (SAR) analysis is listed below.” Below, where? Please specify which Table or figure.
A#8-The phrase "The most important information of structure activity relationship (SAR) analysis is listed below." was changed to "The analysis and interpretation of data presented in Table 1 led to draw the following structure–activity relationships (SAR):"
Q#9-Discussion: Some information about the security of those synthesized molecules could be given.
A#9-Unfortunatly, we did not test the toxicity of synthetized compounds yet.
Q#10-Materials and Methods: The chemical formula should be presented as Figures and numbered accordingly.
A#10-Done.
Reviewer 2 Report
Carbonic anhydrases inhibitors have found pharmacologic applications in many areas. The authors synthesized 12 structurally diverse N-((4-sulfamoylphenyl)carbamothioyl) amides and tested their potential inhibitory effect on CA of different origins. The work is interesting and provides novel insights on the sulfonamide–acyl thiourea derivatives as CA inhibitors. The results also have the potential for pharmaceutical use. For the most part, this work is based on sound designs and experiments.
Manuscript can be further improved taking following points into consideration:
1. The structural information of the six CA should be briefly introduced, since they are the inhibitory target of the chemicals. Based on that, the inhibitory mechanism of the sulfonamide–acyl thiourea derivatives on CAs could be discussed from a structural perspective. I wonder if it is possible to perform a molecular docking between the proteins and the inhibitors. I believe if the above questions could be addressed, this study would be more insightful and helpful for the development of future pharmaceuticals in the relevant field.
Author Response
Comments and Suggestions for Authors
Carbonic anhydrases inhibitors have found pharmacologic applications in many areas. The authors synthesized 12 structurally diverse N-((4-sulfamoylphenyl)carbamothioyl) amides and tested their potential inhibitory effect on CA of different origins. The work is interesting and provides novel insights on the sulfonamide–acyl thiourea derivatives as CA inhibitors. The results also have the potential for pharmaceutical use. For the most part, this work is based on sound designs and experiments.
Manuscript can be further improved taking following points into consideration:
Q#1-The structural information of the six CA should be briefly introduced, since they are the inhibitory target of the chemicals. Based on that, the inhibitory mechanism of the sulfonamide–acyl thiourea derivatives on CAs could be discussed from a structural perspective. I wonder if it is possible to perform a molecular docking between the proteins and the inhibitors. I believe if the above questions could be addressed, this study would be more insightful and helpful for the development of future pharmaceuticals in the relevant field.
A#1-We have performed in silico studies as suggested by the reviewer.
Reviewer 3 Report
The manuscript by C.T. supuran and R. Zalubovskis et al. describes the Inhibition studies on human and mycobacterial carbonic anhydrases with N-((4-sulfamoylphenyl)carbamothioyl)amide derivatives.
12 molecules were synthesized and characterized. They were evaluated against three hCA isoforms and 3 MTB beta-CAs. Inhibition data in the nanomolar range were obtained.
In the introduction, it is not very clear which isoforms are the most interesting to target among all isoforms. Is it possible to have more details?
In Table 1, is the number after the decimal point significant ? Sometimes it is there and sometimes it is not.
Page 3, for a layman, could a reference be associated with AAZ which is used as a control? Why is it a good control ?
A SAR on 12 molecules is not huge. Could docking on these molecules support certain selectivities between isoforms? The SAR is purely descriptive. The authors could have gone further by comparing their molecules with other drugs from the literature for their activity.
These molecules have not been tested on cell lines or mycobacterial strains. Could the authors test them ?
Self-citation is too important in this manuscript: in the section "references", 26 publications are from the same author. Is it justified ?
Author Response
Comments and Suggestions for Authors
The manuscript by C.T. supuran and R. Zalubovskis et al. describes the Inhibition studies on human and mycobacterial carbonic anhydrases with N-((4-sulfamoylphenyl)carbamothioyl)amide derivatives.12 molecules were synthesized and characterized. They were evaluated against three hCA isoforms and 3 MTB beta-CAs. Inhibition data in the nanomolar range were obtained.
Q#1-In the introduction, it is not very clear which isoforms are the most interesting to target among all isoforms. Is it possible to have more details?
A#1-Since all catalytically active isoforms of human carbonic anhydrases have been validated as therapeutic targets for specific disease(s), selective inhibition of each would be desirable.
Q#2-In Table 1, is the number after the decimal point significant? Sometimes it is there and sometimes it is not.
A#2-Done
Q#3-Page 3, for a layman, could a reference be associated with AAZ which is used as a control? Why is it a good control?
A#3-We have introduced the reference of the Stopped-Flow method. About AAZ (acetazolamide), it is a good control because it is a corroborated potent inhibitor of all CA isoforms. Indeed it is clinically used as diuretic for its CA inhibitor properties.
Q#4-A SAR on 12 molecules is not huge. Could docking on these molecules support certain selectivities between isoforms? The SAR is purely descriptive. The authors could have gone further by comparing their molecules with other drugs from the literature for their activity.
A#4-We have performed in silico studies as suggested by the reviewer to explain better the selective profile of the best compounds.
Q#5-These molecules have not been tested on cell lines or mycobacterial strains. Could the authors test them?
A#5-Due to the limited time it is not possible to do such tests. Of note, during the revision of our manuscript we did docking studies.
Q#6-Self-citation is too important in this manuscript: in the section "references", 26 publications are from the same author. Is it justified?
A#6-Since our research group is one of the pioneering groups on the field of drug design of carbonic anhydrase inhibitors, logically the huge number of existed references on the field are related to us.
Round 2
Reviewer 1 Report
Question 1 of my revision was not answered properly. Please read carefully the question and answer all the points asked.
Besides, how the "actual yield" was calculated? how the concentrations were measured to calculate the yields. This must be described in the methodology section.
Section 3: name o the microorganism should be in italics.
Author Response
First, many thanks for noticing that in section 3 still some names of microorganisms are not in italic, that is fixed.
Second, we cannot agree on including of calculations of the "actual yield". I guess by the "actual yield" was meant isolated yield of the compounds. Besides, all the compounds obtained were solids and there were no concentration measurements.